

# IL-4/IL-13 axis as therapeutic targets in allergic rhinitis and asthma

Siti Muhamad Nur Husna[1], Norasnieda Md Shukri[2], Noor Suryani Mohd Ashari[1] and Kah Keng Wong[1]

[1] Department of Immunology, School of Medical Sciences, Universiti Sains Malaysia, Kubang Kerian, Kelantan, Malaysia

[2] Department of Otorhinolaryngology, School of Medical Sciences, Universiti Sains Malaysia, Kubang Kerian, Kelantan, Malaysia

## ABSTRACT

Allergic rhinitis (AR) is a common disorder of the upper airway, while asthma is a disease affecting the lower airway and both diseases are usually comorbid. Interleukin (IL)-4 and IL-13 are critical cytokines in the induction of the pathogenic Th2 responses in AR and asthma. Targeting the IL-4/IL-13 axis at various levels of its signaling pathway has emerged as promising targeted therapy in both AR and asthma patient populations. In this review, we discuss the biological characteristics of IL-4 and IL-13, their signaling pathways, and therapeutic antibodies against each cytokine as well as their receptors. In particular, the pleiotropic roles of IL-4 and IL-13 in orchestrating Th2 responses in AR and asthma patients indicate that dual IL-4/IL-13 blockade is a promising therapeutic strategy for both diseases.

## INTRODUCTION

Allergic rhinitis (AR) is an upper airway disorder due to contact of nasal mucosa with allergens that trigger IgE-mediated inflammation, leading to symptoms such as sneezing, coughing, nasal congestion and rhinorrhea. Asthma, on the other hand, is a lower airway disorder caused by various triggers such as allergies, smoking, air pollution or sinusitis, resulting in airway muscles spasms and inflammation, airway obstruction, wheezing and dyspnea (*Azid et al., 2019*; *Eguiluz-Gracia et al., 2020*; *Meng, Wang & Zhang, 2020*; *Nur Husna et al. 2022a*; *Nur Husna et al. 2021b*; *Sani et al., 2019*). AR and asthma prevalence worldwide has seen a rapid increase where approximately 40% and 334 million people worldwide have been affected with AR and asthma, respectively (*Bousquet et al., 2008*; *Network GAJA, New Zealand, 2014*). Asthma is a fatal disease and accountable for 250,000 possibly preventable deaths annually (*Enilari & Sinha, 2019*). Both AR and asthma usually comorbid with each other in the same individual or in closely related family members, partly due to shared genetic risk variants of immune-related genes in these diseases (*Belsky et al., 2013*; *Ferreira et al., 2017*; *Ober & Yao, 2011*; *Van Beijsterveldt & Boomsma, 2007*). AR is believed to be a risk factor for asthma, with over 80% of asthmatic patients have rhinitis and 10–40% of AR patients have asthma (*Bousquet et al., 2008*; *Bousquet et al., 2001*). The

Corresponding author
Kah Keng Wong, kahkeng@usm.my

"united airway disease hypothesis" proposes that both upper and lower airway diseases are manifestations of an inflammatory process (*Togias, 2003*).

Both diseases are mediated by T helper 2 (Th2) inflammatory processes where Th2 cytokines such as IL-4, IL-5 and IL-13 play key roles in their pathogenesis (*Nur Husna et al. 2022c*). IL-4 plays key roles in inducing IgE production by plasma cells and upregulating the expression of FcεRI and MHC class II molecules in mast cells, basophils, monocytes, macrophages and B cells (*Yamanishi et al., 2017*; *Yip et al., 2021*). IL-4 is considered a master Th2 switch that also drives the generation of other pro-allergic cytokines such as IL-5 and IL-13 by Th2 cells in allergic diseases. IL-4 promotes the development of myeloid dendritic cells (mDCs) and involved in the migration of Th2 cells and eosinophils to the inflamed site. Both IL-4 and IL-13 activate B cells to synthesize IgE, induce goblet cell hyperplasia, trigger airway hyperresponsiveness and mucus hypersecretion (*Sahoo, Wali & Nurieva, 2016*; *Wills-Karp & Finkelman, 2008*). Moreover, Th2 cytokines also reduce the integrity of nasal epithelial barrier in AR and asthma patients by reducing the expression of tight junction (TJ) proteins in epithelial cells (*Heijink, Nawijn & Hackett, 2014*; *Holgate, 2007*; *Steelant et al., 2016b*; *Steelant et al., 2018*).

Our understanding of the pathophysiology of AR has progressed rapidly over the last few decades whereby AR and asthma are exacerbated by aberrantly high cytokines levels (*Gandhi, Pirozzi & Graham, 2017*; *May & Fung, 2015*; *Nur Husna et al. 2022b*; *Yu et al., 2019*) and the disruption of airway epithelial barrier (*Nur Husna et al. 2021a*; *Steelant et al., 2016a*; *Steelant et al., 2018*). However, long-term effective treatments for these diseases are still unmet with rising cases over the years. IL-4/IL-13 axis is important in the pathogenesis of AR and asthma in which the axis exerts a wide range of effects on inflammatory cells and structural changes to airway epithelial cells in AR and asthma. A proportion of chronic AR and asthma patients do not respond towards conventional, symptoms-based medications. These patients require disease-modifying therapeutic agents such as targeted agents against IL-4/IL-13 that can shift the natural course of disease progression in AR and asthma (*Nur Husna et al. 2022c*). Accumulating evidence from *ex vivo* studies as well as AR or asthma animal models lend credence to the further assessment of IL-4/IL-13-targeted therapy in AR and asthma patients (*Bourdin et al., 2021*; *Conde et al., 2021*; *Harb & Chatila, 2020*; *Russkamp et al., 2019*).

Therefore, in this review, the IL-4/IL-13 signaling pathways and therapeutic monoclonal antibodies targeting each cytokine or their receptors, as well as dual IL-4/IL-13 blockade, in both AR and asthma are presented and discussed. The review summarises the key biological and therapeutic features of this vital signaling axis in the two major upper airway disorders that would be informative for otorhinolaryngology clinicians and researchers alike in this field.

## SURVEY METHODOLOGY

This review is pertaining to IL-4/IL-13 axis and their receptors *i.e.,* IL-4Rα, IL-13Rα1 and IL-13Rα2 in AR and asthma research. All articles were searched, retrieved and screened by an investigator (SMNH) and independently verified by another investigator (KKW)

according to the electronic databases Google Scholar and PubMed. Only English language articles were included and articles up to year 2021 were shortlisted. The keywords used were as follows: "allergic rhinitis", "asthma", "AR", "IL-4", "IL-13", "IL-4/IL-13", "IL-4/IL-13 axis", "IL-4Rα", "IL-13Rα1", "IL-13Rα2", "monoclonal antibodies" and "therapeutic antibodies".

## Biological characteristics of IL-4 and IL-13 receptor complexes

The genes encoding IL-4 and IL-13 are located in chromosome 5q31 where this genetic loci is home to a cluster of cytokine genes such as IL-3, IL-4, IL-5, IL-13 and granulocyte-macrophage colony stimulating factor (*Giuffrida et al., 2019*). IL-4 is produced by T cells, basophils and mast cells where it induces differentiation of T cells into Th2 subtype (*Ptackova et al., 2018*). IL-13 is produced by several immune cell populations including Th2 cells, mast cells, B cells, NK cells, innate lymphoid cells and granulocytes (*Giuffrida et al., 2019*; *Joshi et al., 2006*). IL-4 signaling pathway initiates immunoglobulin class switching toward IgE in B cells (*Paul, 2015*), while IL-13 triggers changes in epithelial and smooth muscle cell functions leading to hypersensitivity reactions (*Ito et al., 2009*; *Wills-Karp & Finkelman, 2008*).

IL-4 and IL-13 are structurally similar, multifunctional peptides and share a functional signaling receptor chain. IL-4 binds to two receptors *i.e.,* the type I IL-4 receptor (IL-4R) composed of IL-4Rα and common $\gamma$-chain ($\gamma_c$), and type II IL-4R composed of IL-4Rα and IL-13Rα1. IL-4Rα is present on various types of cells including CD4$^+$ and CD8$^+$ T cells, lung epithelial cells, B cells, macrophages, airway goblet cells, and smooth muscle cells (*Tan, Sugita & Akdis, 2016*). On the other hand, IL-13R $\alpha$1 is expressed by B cells, eosinophils, macrophages, lung epithelial cells, airway goblet cells, and endothelial cells (*Tan, Sugita & Akdis, 2016*).

The type I IL-4R complex is found on the surface of lymphoid T and NK cells, basophils, mast cells and most mouse B cells, while type II IL-4R is present on the surface of non-lymphoid and tumor cells (*Koller et al., 2010*; *Wills-Karp & Finkelman, 2008*). Another receptor that binds IL-13 only is IL-13Rα2 where it acts as a decoy receptor, and usually overexpressed as well as activated in tumor cells or fibrosis (*Bartolome, Jaen & Casal, 2018*; *Fichtner-Feigl et al., 2006*).

## IL-4 and IL-13 activate the IL-4R$\alpha$/STAT6 pathway to induce allergic responses

Allergic response is triggered when DCs present allergen peptides to CD4$^+$ T cells that then differentiate into Th2 cells to produce Th2 cytokines. Binding of IL-4 with its type I receptor which comprises of IL-4Rα and $\gamma_c$ chain leads to activation of JAK1 and JAK3, respectively. For type II IL-4R, binding of IL-13 with IL-13Rα1 subunit activates TYK2/JAK2 (*Nguyen et al., 2020*; *Wang et al., 2020*). Activated JAKs facilitate tyrosine residues phosphorylation of the cytoplasmic tail of IL-4R. Following JAK3 activation, inflammatory mediators (*e.g.*, histamines, leukotrienes) and cytokines (*e.g.*, IL-4, IL-13 and IL-9) are released which influence the responses of neighboring cells (*i.e* B cells, mast cells, macrophages, dendritic cells and endothelial cells), triggering IgE production by

plasma cells, eosinophils infiltration, airway inflammation, bronchoconstriction and tissue damage (*Malaviya, Laskin & Malaviya, 2010*).

The phosphorylated tyrosine residues serve as docking sites for STAT6 (a transcription factor selectively coupled to the IL-4Rα chain), activating IL-4 and IL-13 responsive genes in the subsequent signaling pathway of allergic responses. STAT6 induces Sonic hedgehog expression in the airway epithelium leading to goblet cell metaplasia and enhanced mucus production in asthma. pSTAT6 is associated with the production of Th9 cells and IL-9 during airway inflammation (*Hoppenot et al., 2015*). Activated macrophage marker Arginase 1 (Arg-1) can be induced by IL-4 and IL-13 which ultimately increase the production of L-ornithine and its downstream products polyamines and L-proline (*i.e.,* inflammatory mediators causing airway remodeling) (*Van Den Berg, Meurs & Gosens, 2018*). Arg-1 inhibitors were recently patented for the treatment of AR and asthma (*Meurs et al., 2019*).

This activates B cells to produce circulating IgE that binds to specific Fcε receptors on mast cells and basophils. Signaling activation from Th2-type cytokines are important survival signals for mast cells, basophils, and eosinophils. Degranulation of mast cells results in the release of inflammatory mediators such as histamine, tryptase, chymase, kininogenase (generates bradykinin), heparin, prostaglandin D2 and the sulfidopeptidyl leukotrienes (*Skoner, 2001*). In AR, these mediators induce mucosal edema and watery rhinorrhea, while histamine activates its H1 receptors on sensory nerve endings that causes sneezing, pruritus, and reflex secretory responses. Moreover, histamine-mediated activation of H1 and H2 receptors on mucosal blood vessels leads to nasal congestion and plasma leakage (*Sin & Togias, 2011*). During the late phase in AR, nasal mucosal inflammation occurs with the influx and activation of a variety of inflammatory cells (*i.e.,* T cells, eosinophils, basophils, neutrophils, and monocytes) into nasal mucosa that depends on Th2 cytokines *e.g.*, IL-4, IL-5 and IL-13 (*Sin & Togias, 2011*). In asthma, the provocative discharges can spread from the upper airway to lungs, which triggers inflammation and bronchoconstriction (*Stone, Prussin & Metcalfe, 2010*). Airway hyperresponsiveness is a hallmark feature of asthma due to exaggerated bronchoconstrictor response (*Sinyor & Perez, 2020*).

Furthermore, IL-4 and IL-13 are not only responsible for initiating inflammatory responses, but they also play important roles in disrupting nasal epithelial barrier integrity. The cytokines may accumulate within the sinonasal microenvironment surrounding sinonasal epithelial cells, activating IL-4/IL-13 axis signaling cascade that impairs epithelial TJ composition (*Capaldo & Nusrat, 2009*; *London Jr, Tharakan & Ramanathan Jr, 2016*). The signaling cascade deregulates the expression and assembly of epithelial TJ molecules, leading to increased nasal epithelial permeability to allergens (*Capaldo & Nusrat, 2009*; *Steelant et al., 2018*). The cytokines also obstruct the epithelial barrier from resealing which preserves the contact with inflammatory allergens (*London Jr, Tharakan & Ramanathan Jr, 2016*).

Association of IL-4/IL-13 axis with impaired nasal epithelial barrier integrity has been observed in both AR and asthma patients. In AR patients, treatment with anti-IL-4Rα monoclonal antibodies (mAbs) successfully restored epithelial barrier integrity and function (*Steelant et al., 2018*). IL-4 could disrupt epithelial integrity of primary nasal

epithelial cells by reducing TJ molecules expression including zonula occludens-1 (*ZO1*) and occludin (*OCLN*) (*Steelant et al., 2016a*). Reduced expression of these TJ molecules in nasal epithelial cells are frequently observed in AR patients (*Nur Husna et al., 2021a*; *Siti Sarah et al., 2020*; *Steelant et al., 2016a*; *Wang Ms et al., 2021*). Addition of IL-4 and IL-13 to reconstructed human epidermis (RHE) cells resulted in downregulation of *CLDN1* expression in AR (*Gruber et al., 2015*).

The IL-4/IL-13 axis also plays important roles in the pathophysiology of inflammatory arthritis especially rheumatoid arthritis (RA). In RA, IL-4/IL-13 cytokines promote the production of proinflammatory cytokines such as IL-1β and TNF-α, as well as macrophage polarization from classically activated (M1) phenotype into the alternatively activated (M2) phenotype (*Iwaszko, Biały & Bogunia-Kubik, 2021*). These collectively promote inflammation of the joints in RA patients. On the other hand, in AR patients, IL-4/IL-13 axis primarily induces the recruitment of mDCs, eosinophils and overproduction of IgE by plasma cells, as well as repressing the expression of TJs by nasal epithelial cells. The pathophysiological differences of IL-4/IL-13 axis in RA and AR patients are likely due to distinct causative factors in both diseases whereby in AR, allergens are the key trigger while RA is multifactorial (*i.e.,* hormones, genetics and environmental factors), leading to dissimilar clinical manifestations.

IL-4/IL-13 axis is a promising therapeutic target in AR and asthma. Development of novel therapies targeting IL-4/IL-13 axis can be achieved by disrupting its signaling axis at various molecular levels such as inhibition of the soluble cytokines and their receptors on cell surfaces. Several therapeutic mAbs targeting IL-4 and IL-13 and their receptors have been developed to date and are discussed in the next sections.

## Therapeutic antibodies targeting IL-4/IL-13 axis in AR
### Anti-IL-4 mAb in AR
VAK694 is a fully humanized anti-IL-4 mAb that has been assessed in AR patients. A phase IIa randomized controlled trial (RCT) of VAK694 in AR patients to examine its efficacy had been conducted (*Chaker et al., 2016*). The VAK694 dosage (3 mg/kg every 4 weeks) used was based on previous phase I studies (NCT00620230 and NCT00929968). In this study, seasonal AR patients ($n = 37$) were randomized into three groups *i.e.,* combination of subcutaneous allergen immunotherapy (SCIT) with VAK694, suboptimal SCIT with placebo antibody, and double placebo (placebo SCIT and placebo antibody) and the primary endpoint assessed was suppression of skin late-phase response (LPR). However, no significant difference in the skin LPR in the VAK694 plus SCIT group compared with the SCIT-only group. The authors concluded that anti–IL-4 treatment with VAK694 was effective in modulating Th2 memory when applied during administration of specific immunotherapy, but it did not provide additional clinical efficacy. It was further suggested that upcoming clinical studies to include combined intervention with allergen, allergen dose, and biological agents.

### Anti-IL-13 mAb in AR
Dectrekumab (also known as QAX-576) is a fully human investigational anti-IL-13 mAb with anti-inflammatory potential. It is widely investigated in immune disorders including

AR asthma, eosinophilic esophagitis (EoE), Crohn's disease and keloids (*Rothenberg et al., 2015*). A phase II double-blind study on dectrekumab (QAX576) aimed to evaluate the effects of the anti–IL-13 mAb ($n = 16$) compared with placebo ($n = 15$) on repeated nasal allergen challenge responses in AR patients out of season (*Kariyawasam et al., 2009*; *Nicholson et al., 2011*).

A significant decrease in IL-13 levels was observed in patients administered with anti–IL-13 mAb compared with the placebo group after nasal allergen challenge on day 5 and day 7. However, there were no obvious effects of dectrekumab on nasal lavage eosinophil numbers or total nasal symptom scores compared to placebo. Dectrekumab treatment only managed to decrease total nasal symptom scores in a subgroup with high late-phase nasal IL-13 levels. The antibody inhibited nasal IL-13 responses but failed to inhibit nasal symptoms and eosinophils, suggesting that anti-IL-13 is not critical for acute nasal allergic response (*Kariyawasam et al., 2009*).

## Therapeutic antibodies targeting IL-4/IL-13 axis in asthma
### Anti-IL-4 mAb in Asthma
Pascolizumab (SB 240683) is a humanized mAb against IL-4. A preclinical study was first conducted to generate pascolizumab as a murine mAb (3B9) with specificity for human IL-4, which was subsequently humanized (*Hart et al., 2002*). It was tested in cynomolgus monkeys where they received monthly intravenous doses (up to 100 mg per kg) for 9 months, and pascolizumab was reported to be well-tolerated (*Hart et al., 2002*). The dose-escalation phase I RCT of pascolizumab was assessed in mild-to-moderate asthma adult patients. Pascolizumab was well-tolerated at single intravenous doses of 0.5–10 mg/kg with an elimination half-life of more than 2 weeks (*Shames et al., 2001*). A multidose phase I/II trial of pascolizumab in symptomatic steroid-naïve subjects with asthma ($n = 120$) was conducted, however it was terminated because preliminary data showed that pascolizumab did not provide clinical benefit (NCT00024544).

### Anti-IL-13 mAb in Asthma
Numerous experimental studies have shown that IL-13 as a potential therapeutic target in asthma. In mouse models of asthma, intravenous IL-13-targeted mAb administration effectively inhibited allergen-induced inflammation, goblet cell hyperplasia and airway remodelling (*Yang et al., 2004*). Lebrikizumab, a humanized anti-IL-13 mAb, was produced to potentially treat asthma patients with "Th2-high" asthmatic phenotypes characterized by overexpression of IL-13-inducible genes such as the gene encoding periostin (an extracellular matrix protein produced by bronchial epithelial cells) (*Woodruff et al., 2009*). Periostin plays pathogenic roles in asthma patients where it induces mucus secretion, airway fibrosis and tissue remodelling (*O'Dwyer & Moore, 2017*).

In a randomized multicenter study (*Corren et al., 2011*) of moderate-to-severe asthmatic adult patients ($n = 219$) whose disease was inadequately controlled by inhaled corticosteroid therapy, lebrikizumab was administered at monthly subcutaneous doses of 250 mg for 6 months. Lebrikizumab elicited improved lung function in these patients who demonstrated high serum levels of periostin. At week 12, the forced expiratory volume in the first second (FEV1) increased compared with baseline values for lebrikizumab-treated group, and
this was more pronounced in the high-periostin than low-periostin subgroup of patients. In the LUTE ($n = 258$) and VERSE ($n = 205$) replicated phase IIb RCTs, the efficacy of lebrikizumab as anti-IL-13 mAb was assessed in moderate-to-severe asthma patients (*Hanania et al., 2015*). Treatment with lebrikizumab resulted in 60% reduction in the rate of asthma exacerbations, which was more noticeable in the periostin-high patients compared with the periostin-low patients. Improvement in FEV1 was also higher in periostin-high patients compared with periostin-low patients. Collectively, these studies demonstrated the significant efficacy of lebrikizumab in periostin-high patients.

In phase III trials of LAVOLTA I ($n = 1,081$) and LAVOLTA II ($n = 1,067$), the efficacy and safety of lebrikizumab in adult patients with uncontrolled asthma, despite inhaled corticosteroids and at least one-second controller medication, was assessed (*Hanania et al., 2016*). Patients were randomly assigned to receive lebrikizumab or placebo subcutaneously, once every 4 weeks. It was shown that the exacerbation rates were reduced in biomarker-high patients (periostin $\geq$ 50 ng/mL or blood eosinophils $\geq$ 300 cells per $\mu$L) treated with lebrikizumab *versus* placebo in both trials. Lebrikizumab successfully blocked IL-13, however it did not consistently show significant reduction in asthma exacerbations in biomarker-high patients.

Tralokinumab is a fully human IgG4 mAb that employs a distinct mode of action from lebrikizumab where it binds to IL-13 cytokine at an epitope that overlaps with the binding site of the IL-13R$\alpha$ receptors, preventing IL-13 from binding to both IL-13R$\alpha$1 and IL-13R$\alpha$2 (*Popovic et al., 2017*). Interestingly, this strategy is proposed to be more efficacious than blocking the binding to IL-13R$\alpha$1 alone (*Tripp et al., 2017*). Preclinical studies of tralokinumab in mice model of respiratory and esophageal inflammation induced by intratracheal human IL-13 showed that the mAb markedly attenuated airway eosinophilia and bronchial hyperresponsiveness (*Blanchard et al., 2005*). In the two phase I clinical trials of tralokinumab, this antibody demonstrated an acceptable safety profile in asthmatic patients ($n = 23$) (*Singh et al., 2010*) and administration of 300 mg tralokinumab was well tolerated in asthma patients ($n = 20$) in another trial (*Baverel et al., 2015*).

Phase II trial was conducted in moderate-to-severe uncontrolled asthma ($n = 194$) resulted in improved lung function (FEV1) compared to placebo, however no changes in Asthma Control Questionnaire score (ACQ) was observed (*Piper et al., 2013*). A non-significant reduction in asthma exacerbation rates was observed in severe uncontrolled asthma patients ($n = 150$) in a separate phase IIb trial (*Brightling et al., 2015*). Additionally, eosinophil count in bronchial biopsy was assessed in phase II trial (MESOS) of inadequately controlled moderate-to-severe asthma patients ($n = 39$) *versus* placebo demonstrated insignificant changes in bronchial eosinophil count, however tralokinumab reduced fractional exhaled nitric oxide (FeNO) and total blood IgE concentrations in the patients (*Russell et al., 2018*).

In two phase III trials, STRATOS 1 (all-comers population, $n = 398$) and STRATOS 2 (FeNO-high population, $n = 108$), both trials was assessed in severe, uncontrolled asthma patients (*Panettieri Jr et al., 2018*). In the STRATOS 1 study, tralokinumab reduced asthma exacerbation rate in severe asthma patients (FeNO $\geq$ 37 ppb), but the reduction was insignificant in the STRATOS 2 study. These inconsistent results suggested that IL-13

signaling pathway may not play causative roles of severe asthma exacerbation. Lastly, phase III trial (TROPOS) of tralokinumab in severe, uncontrolled asthma patients ($n = 140$) aimed to reduce the dependency of oral corticosteroid (OCS) use. However, no significant changes were observed in the final daily average OCS dose compared to placebo (*Busse et al., 2019*). Taken together, tralokinumab administration has not yielded clinical benefits in controlling asthma.

## Dupilumab as a dual IL-4/IL-13 blockade antibody in AR and asthma

The recent failure of the IL-13 inhibitor tralokinumab to show efficacy in phase III trials of asthma patients underscores the need for other more effective IL-4/IL-13 inhibitors. Dupixent (dupilumab) is a fully human mAb (*Macdonald et al., 2014*). It is a first-in-class biologic that binds to both monomeric and dimeric human IL-4Rα (type I and type II receptors), with $K_D$ values of 33 pM and 12 pM, respectively, preventing both IL-4- and IL-13-mediated signaling pathways. The antibody has demonstrated clinical benefits for patients with type 2 signature diseases, leading to its approval for the treatment of asthma, chronic sinusitis patients with nasal polyposis, and moderate-to-severe atopic dermatitis (*Beck et al., 2014*; *Blauvelt et al., 2017*; *Hamilton et al., 2014*; *Le Floc'h et al., 2020*; *Rabe et al., 2018*; *Simpson et al., 2016*; *Thaci et al., 2016*; *Wenzel et al., 2013*).

In preclinical investigation utilizing primary cell assays and murine model of house dust mite (HDM)–induced asthma, the efficacy of IL-4 *vs* IL-13 *vs* IL-4Rα blockers was assessed (*Le Floc'h et al., 2020*). Dupilumab was responsible to reduce thymus and activation-regulated chemokine (TARC) and IgE levels (*i.e.,* two effector molecules implicated in type 2 diseases), and fully blocked CD23 upregulation on B cells and IL-12p70 secretion from monocyte-derived DCs (*Le Floc'h et al., 2020*). Dupilumab also suppressed the infiltration of pathogenic cells such as ST2$^+$CD4$^+$ T cells (*i.e.,* key cells that express the IL-33 receptor, ST2, IL-5 and IL-13 during allergic inflammation) and eosinophils, as well as repressing activated B cells both locally and systemically (*Endo et al., 2014*; *Mato et al., 2017*; *Wambre et al., 2017*). Moreover, dupilumab also inhibited HDM-induced expression of CCL26 (eotaxin 3), CCL2 (MCP-1), IL-6 and goblet cell metaplasia responsible for mucus production (*Le Floc'h et al., 2020*). Strikingly, treatment of HDM-exposed mice with dupilumab demonstrated an improvement in lung functions (FEV1) (*Gandhi et al., 2016*; *Gandhi, Pirozzi & Graham, 2017*).

In clinical trials, dupilumab therapy was able to improve lung functions and reduced levels of Th2-driven inflammatory markers (*e.g.,* TARC, eotaxin-3, and IgE) compared to placebo group in phase II clinical trial of dupilumab in patients with persistent, moderate-to-severe asthma with increased levels of eosinophil ($n = 52$) (*Wenzel et al., 2013*). A greater benefit was observed in patients with higher baseline levels of eosinophil in phase III clinical trial (Liberty Asthma QUEST). In the study, moderate-to-severe asthma patients ($n = 1,902$) were randomly assigned to receive dupilumab (dose of 200 or 300 mg every two weeks) ($n = 631$) or matched-volume placebos ($n = 317$) for 52 weeks (*Castro et al., 2018*). Significantly lower rates of severe asthma exacerbation and improved FEV1 were observed in patients who received dupilumab compared to placebo. Patients with higher baseline

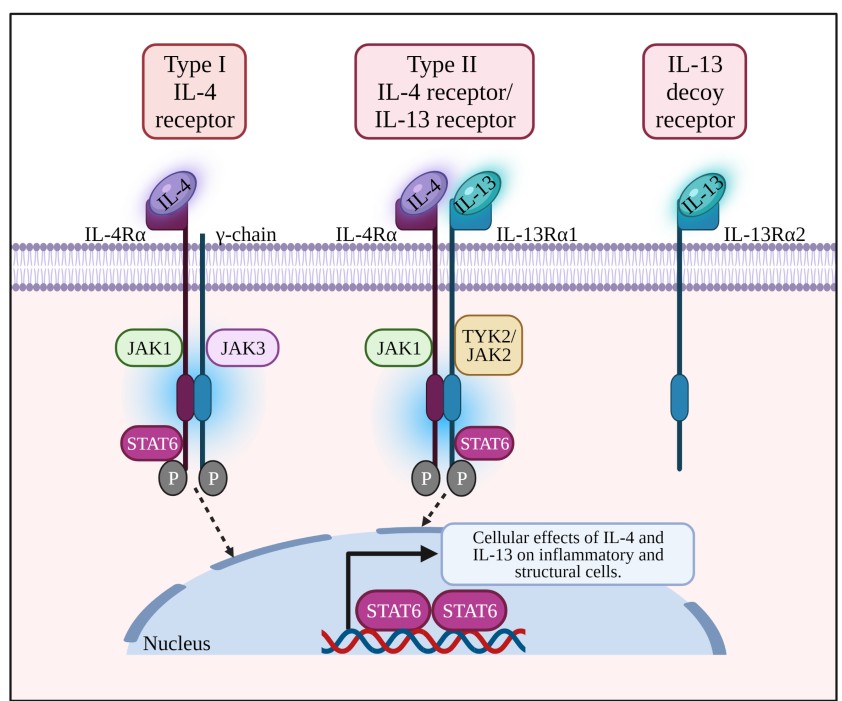

**Figure 1 IL-4 binds to type I receptor (composed of IL-4Rα and common γ-chain) and type II receptor (composed of IL-4Rα and IL-13Rα1). IL-13 binds to type II receptor, and IL-13Rα2 acts as a decoy receptor.** Binding of a ligand by type I or II receptor results in the activation of Janus family kinases (JAK1, JAK2/ TYK2 and JAK3) followed by the phosphorylation of a signal transducer and activator transcription 6 (STAT6). IL-13: Interleukin-13; IL-13R α1:Interleukin-13 receptor alpha 1; IL-13R α2:Interleukin-13 receptor alpha 2; IL-4: Interleukin-4; IL-4R α: Interleukin-4 receptor alpha; IL-5: Interleukin-5; IL-6: Interleukin-6; JAK1: Janus kinase 1; JAK2: Janus kinase 2; JAK3: Janus kinase 3; P: Phosphate; STAT6: Signal transducer and activator of transcription 6; Th2: T helper 2; TYK2: Tyrosine kinase 2.

levels of eosinophils exhibited better responses in terms of both exacerbations and the FEV1.

In an accompanying trial, assessment of dupilumab efficacy *versus* placebo was conducted in a phase III trial (LIBERTY ASTHMA VENTURE) in oral glucocorticoid-dependent severe asthma patients ($n = 210$) (*Rabe et al., 2018*). Add-on therapy with dupilumab significantly reduced OCS dose used and severe exacerbation rate together with improved FEV1 in the dupilumab group. The first real life cohort study (DUPI-France) (*Dupin et al., 2020*) on dupilumab in steroid-dependent severe asthmatic patients ($n = 64$) resulted in improved asthma control, lung functions and reduced oral steroids use. Further improvement in the lung function from these three studies showed a potential effect of dupilumab in airway remodeling. In the most recent phase III clinical trial (TRAVERSE) of dupilumab (*Wechsler et al., 2021*), the long-term safety and efficacy of dupilumab was assessed in moderate-to-severe or oral corticosteroid-dependent severe asthma ($n = 2,282$). Long-term safety profile in terms of severe asthma exacerbations and lung functions were achieved. The IL-4/IL-13 axis signaling pathway is summarized in Fig. 1. Schematic representation of the cellular effects of IL-4/IL-13

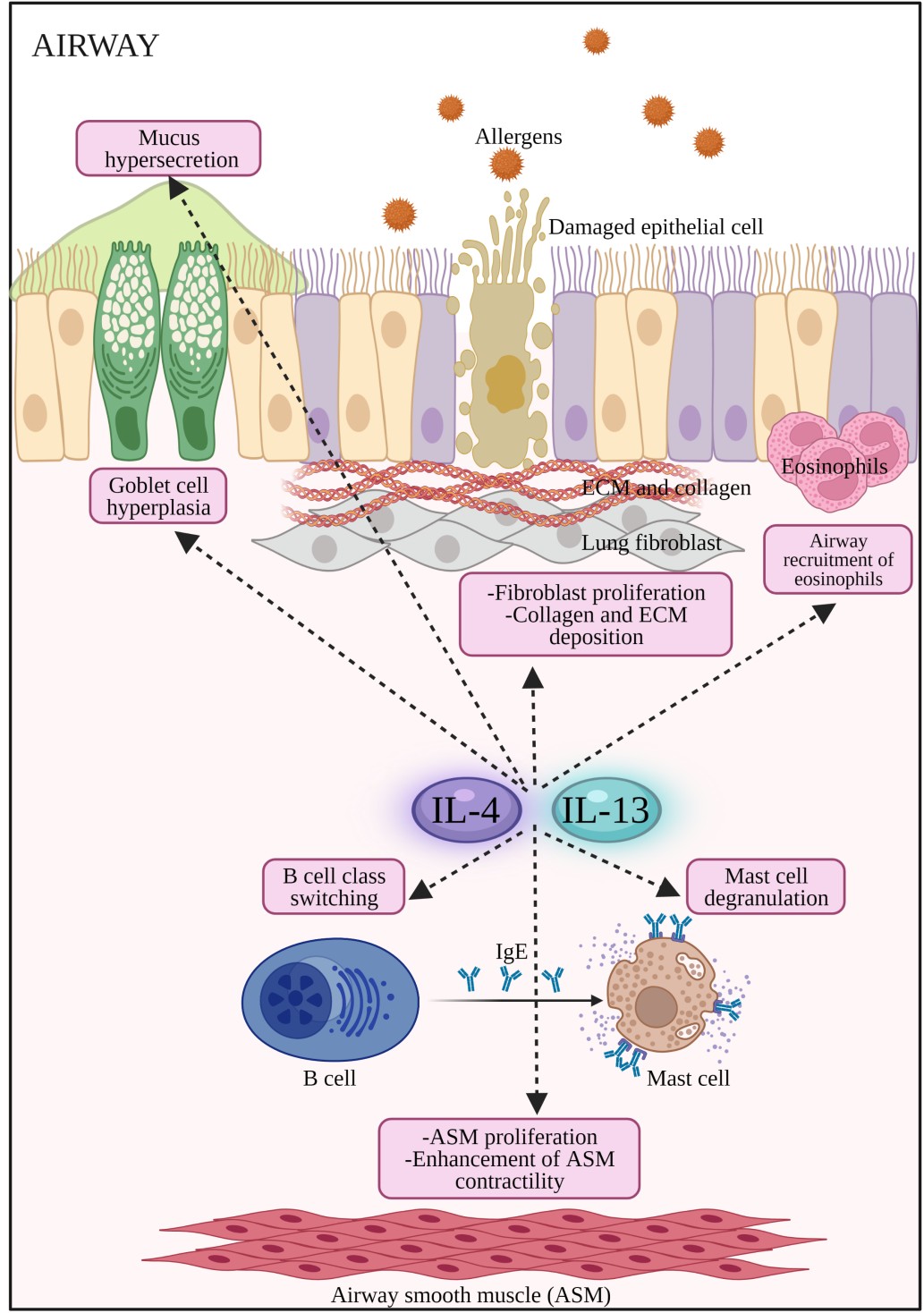

**Figure 2  The potential effects of IL-4 and IL-13 on inflammatory cells and structural changes to epithelial barrier in AR and asthma.** Both IL-4 and IL-13 promote B cell class switching to produce IgE, mast cell degranulation to release inflammatory mediators such as histamines, (continued on next page…)

**Figure 2 (…continued)**
airway inflammation by the recruitment of eosinophils, promote airway hyperresponsiveness by stimulating goblet cell hyperplasia, mucus hypersecretion, proliferation of fibroblast, collagen and ECM deposition, as well as proliferation and enhancement of ASM contractility. ASM: Airway smooth muscle; ECM: extracellular matrix; IL-13: Interleukin-13; IL-4: Interleukin-4.

axis in AR and asthma are summarized in Fig. 2. In addition, completed and ongoing clinical trials of anti-IL-4/IL-13 axis mAbs in AR and asthma patients are presented in Table 1.

## CONCLUSION

Therapies targeting IL-4/IL-13 axis serve as disease-modifying agents that can restore the immune system's homeostasis. Clinical trials targeting either IL-4 or IL-13 in AR have not yielded beneficial clinical outcomes. Trials involving lebrikizumab and tralokinumab (anti-IL-13 mAb) demonstrated improvements in lung functions but not in asthma symptoms. Hence, targeting both IL-4 and IL-13 has proven to be more successful as exemplified by dupilumab that could reduce inflammatory-driven mediators (*e.g.*, serum IgE, eotaxin-3, TARC and exhaled nitric oxide). However, dupilumab clinical trials do not represent a 'real world' asthma with short study duration, and longer duration of dupilumab clinical trials are warranted to assess the long-term safety and tolerability profiles of dupilumab.

Nur Husna et al. (2022), *PeerJ*, DOI 10.7717/peerj.13444

**Table 1** Completed and ongoing clinical trials of therapeutic antibodies targeting IL-4/IL-13 axis and their receptors in AR and asthma patients.

| Disease | Target | Drug name | Status of clinical trial | Clinical trials identifier number | Phase | Patient population |
|---|---|---|---|---|---|---|
| **Allergic rhinitis** | IL-4 | VAK694 | Completed | NCT00929968 | I | Atopic subjects with seasonal rhinitis ($n = 35$) |
| | | | | NCT00620230 | I | Patients with atopic disease ($n = 46$) |
| | | | | NCT01018693 (*Chaker et al., 2016*) | IIa | Seasonal AR patients ($n = 37$) |
| | IL-13 | Dectrekumab | Completed | NCT00584584 (*Nicholson et al., 2011*) | II | AR patients ($n = 16$) |
| | IL-4Rα | Dupilumab | Recruiting/Ongoing | NCT04502966 (GRADUATE) | II | Moderate to severe seasonal AR and allergic sensitization to grass pollen ($n = 108$) |
| **Asthma** | IL-4 | Pascolizumab (SB 240683) | Completed | NCT00024544 | I/II | Symptomatic steroid-naïve subjects with asthma patients ($n = 120$) |
| | IL-4Rα | Altrakincept | Completed | N/A (*Borish et al., 1999*) | I/II | Moderate atopic asthma patients ($n = 25$) |
| | | | | NCT00001909 (*Borish et al., 2001*) | II | Adult asthma patients ($n = 40$) |
| | | AMG 317 | Completed | NCT00436670 (*Corren et al., 2010*) | II | Moderate to severe asthma patients ($n = 294$) |
| | | Pitrakinra | Completed | NCT00535028 (*Wenzel et al., 2007*) | IIa | Allergen challenged asthmatic patients ($n = 24$) |
| | | | | NCT00535031 (*Wenzel et al., 2007*) | IIa | A Allergen challenged asthmatic patients ($n = 32$) |
| | | | | NCT00801853 (*Slager et al., 2012*) | II | Asthma patients ($n = 424$) |
| | | Dupilumab | Completed | NCT01312961 (*Corren et al., 2010*) | II | Persistent Moderate to Severe Eosinophilic Asthma |
| | | | | NCT02528214 (LIBERTY ASTHMA VENTURE) (*Rabe et al., 2018*) | III | Oral glucocorticoid-dependent severe asthma ($n = 210$) |
| | | | | NCT02414854 (LIBERTY ASTHMA QUEST) (*Castro et al., 2018*) | III | Moderate-to-severe asthma patients ($n = 1902$) |
| | | | | NCT04022447 (DUPI-France) (*Dupin et al., 2020*) | Real-life setting | Steroid dependent severe asthmatic patients ($n = 642$) |
| | | | | NCT02134028 (LIBERTY ASTHMA TRANVERSE) (*Wechsler et al., 2021*) | III | Moderate-to-severe or oral corticosteroid (OCS)-dependent severe asthma ($n = 2282$) |

Nur Husna et al. (2022), *PeerJ*, DOI 10.7717/peerj.13444

**Table 1** (*continued*)

| Disease | Target | Drug name | Status of clinical trial | Clinical trials identifier number | Phase | Patient population |
|---------|--------|-----------|--------------------------|-----------------------------------|-------|---------------------|
| | | Dupilumab | Recruiting/Ongoing | NCT03694158 | IV | Adolescents and adults with asthma ($n = 126$) |
| | | | | NCT04287621 | Real-life setting | Asthma patients ($n = 1000$) |
| | IL-13 | Cendakimab (RPC4046) | Completed | NCT00986037 (*Tripp et al., 2017*) | I | Mild to moderate controlled asthma patients ($n = 27$) |
| | | Lebrikizumab | Completed | NCT01545440 (LUTE) (*Hanania et al., 2015*) | IIb | Moderate-to-severe asthma patients ($n = 258$) |
| | | | | NCT01545453 (VERSE) (*Hanania et al., 2015*) | IIb | Moderate-to-severe asthma patients ($n = 205$) |
| | | | | NCT01867125 (LOVALTA I) (*Hanania et al., 2016*) | III | Uncontrolled asthma patients ($n = 1081$) |
| | | | | NCT01868061 (LAVOLTA II) (*Hanania et al., 2016*) | III | Uncontrolled asthma patients ($n = 1067$) |
| | | Tralokinumab | Completed | NCT00974675 (*Singh et al., 2010*) | I | Asthma patients ($n = 23$) |
| | | | | NCT01592396 (*Baverel et al., 2015*) | I | Asthma patients ($n = 23$) |
| | | | | NCT00873860 (*Piper et al., 2013*) | II | Moderate-to-severe uncontrolled asthma ($n = 194$) |
| | | | | NCT01402986 (*Brightling et al., 2015*) | IIb | Severe uncontrolled asthma patients ($n = 150$) |
| | | | | NCT02449473 (MESOS) (*Russell et al., 2018*) | II | Moderate-to-severe asthma patients ($n = 39$) |
| | | | | NCT02161757 (STRATOS 1) (*Panettieri Jr et al., 2018*) | III | Severe, uncontrolled asthma ($n = 398$) |
| | | | | NCT02194699 (STRATOS 2) (*Panettieri Jr et al., 2018*) | III | Severe, uncontrolled asthma ($n = 108$) |
| | | | | NCT02281357 (TROPOS) (*Busse et al., 2019*) | III | Severe, uncontrolled asthma patients ($n = 140$) |
| | | IMA-638 and IMA-026 | Completed | NCT00410280 & NCT00725582 (*Gauvreau et al., 2011*) | I/II | Mild atopic asthma ($n = 56$) |

### Funding

We received funding from the Universiti Sains Malaysia comprising the Research University Grant (1001/PPSP/8012349) awarded to Kah Keng Wong and the Research University Grant (1001.PPSP.8012285) awarded to Noor Suryani Mohd Ashari. The funders had no role in study design, data collection and analysis, decision to publish, or preparation of the manuscript.

### Grant Disclosures

The following grant information was disclosed by the authors:
The Universiti Sains Malaysia comprising the Research University Grant: 1001/PP-SP/8012349.
The Research University Grant: 1001.PPSP.8012285.

### Competing Interests

The authors declare there are no competing interests.

### Author Contributions

- Siti Muhamad Nur Husna conceived, designed and wrote the manuscript, performed literature search, prepared the figures and tables, revised the manuscript and approved the final draft.
- Norasnieda Md Shukri conceived and designed the experiments, authored or reviewed drafts of the paper, and approved the final draft.
- Noor Suryani Mohd Ashari conceived and designed the experiments, authored or reviewed drafts of the paper, and approved the final draft.
- Kah Keng Wong conceived, designed and wrote the manuscript, performed literature search, revised the manuscript, and approved the final draft.

### Data Availability

   This is a review article.

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
