# Peer review of "IL-4/IL-13 axis as therapeutic targets in allergic rhinitis and asthma"

_PeerJ, doi:10.7717/peerj.13444_

## Round 0.1 · original submission · Major Revisions

Dear Authors,

Please revise your manuscript according to reviewers' comments or write a detailed rebuttal on a point-by-point basis.

·

Basic reporting

Yes the review if of broad and cross-disciplinary interest. The topic and content discussed in this manuscript is within the scope of the journal

oThe role of IL-4/IL-13 in allergic response has not been reviewed. This manuscript provides novel insights and informative summaries of recent development in this field. From a quick pubmed search, another IL-4 / IL-13 review (published on Cells in 2021) that might be of interest describes their involvement on inflammatory arthritis with a specific focus on RA. (Link: https://pubmed.ncbi.nlm.nih.gov/34831223/) It would be interesting to compare and contrast some of the pathophysiological pathways in AR/asthma and RA as IL-4/IL-13’s involvement in inflammatory arthritis is well-studied.

The Introduction provided concise context on the pathophysiology of AR and asthma as well as the involvement of IL-4 and IL-13 in this process. However, it failed to mention what is missing / unknown in this field that this manuscript can provide valuable insights on. It would be great if the authors can provide their opinions on where the future research should focus on to better enhance our understanding of the Il-4/IL-13 signaling pathways, as well as clinical research into AR and asthma. The transition from describing the signaling pathways to therapeutic antibodies targeting IL-4/IL-13 is very abrupt. It would be great if the authors can add a few sentences on why IL-4 and IL-13 have been used as therapeutic target for AR and asthma to better establish this link.

Experimental design

Yes the survey methodology is consistent with a comprehensive and unbiased coverage of the subject.

The authors did a great job sourcing / citing relevant publications. The summary table of clinical studies targeting IL-4/IL-13 for AR and asthma is very helpful.

The organization and subsections are also appropriate. The manuscript is structured and presented in a reader friendly manner.

Validity of the findings

Overall comments: the authors provided very good descriptions and summaries of clinical studies targeting the IL-4/IL-13 axis for AR and asthma. The authors included studies with both positive and negative results on the therapeutic potential of IL-4/IL-13 targeted agents. However, the authors spent a lot of texts summarizing the past / recent clinical studies targeting IL-4/IL-13, but the descriptions on the biochemical and cellular pathways regulated by IL-4 and IL-13 are disproportionally limited. Figure 1 provides great illustration on the complex signaling pathway, but it would be great if the authors can elaborate more on the involvement of IL-4/IL-13 in AR and asthma.

Row 122-124 "IL-4Rα/13Rα1 high-affinity heterodimer receptor has the ability to bind both IL-4 and IL-13, which clarifies the similarity of their biological functions in the development of allergic responses."
This is an overstatement. Just because the IL-4Ra/13Ra1 heterodimer receptor has the ability to bind both IL-4 and IL-13, it is not sufficient to draw the conclusions that IL-4 and IL-13 has similar biological functions during allergic responses.The authors should provide additional literature to support this statement or additional rationales / evidences to demonstrate that IL-4 and IL-13 truly play similar functions in allergic responses


The Conclusion section is week and contains statements that are not accurate. Specifically, the authors only mentioned the clinical success of dupilumab in AR/asthma but neglected the many trials with negative / inconclusive results. It would be better if the authors can provide an unbiased and less rosy view on the clinical outlook of IL-4/IL-13 targeted therapeutics. Given the various trial results, it would also be worth discussing what might have caused the different outcomes and potential hypothesis on the limitations of only targeting the IL-4/IL-13 axis

Additional comments

N/A

·

Basic reporting

The paper is well structured and written, with clear objectives. The topic is an important subject.

Experimental design

This review is organized logically into coherent paragraphs/subsections.

Validity of the findings

In this manuscript, there are well-developed and supported arguments that meet the goals set out in the Introduction, and the authors also discuss the future directions in in AR and asthma.

Additional comments

In this review, the authors demonstrated that the type II cytokines IL-4 and IL-13 play an important role in the pathogenesis of allergic rhinitis and asthma. The authors also discussed the biological characteristics of IL-4 and IL-13, their signaling pathways, and therapeutic monoclonal antibodies targeting each cytokine as well as their receptors in both AR and asthma. Then the authors suggested that the pleiotropic roles of IL-4 and IL-13 in orchestrating Th2 responses in AR and asthma patients indicate that dual IL-4/IL-13 blockade is a promising therapeutic strategy for both diseases. However, there are several concerns that the authors need to address/clarify further and validate the current conclusions.

1. The authors need to add more epidemiological information in the introduction such as how many people do AR and asthma affect, and the fatality rate.
2. I suggest removing T cell, eosinophil, basophil, neutrophil, monocyte, RANTES, and MUC5AC in the Figure. It's not very appropriate to put them in a nucleus.
3. The authors show that the IL-4/IL-13 signaling pathway plays an important role in both AR and asthma, and those therapeutic antibodies against each cytokine have emerged as promising targeted therapies for AR and asthma. It is recommended that briefly introduce the role of other factors in this signaling pathway in asthma. such as STAT6, JAK3, or Arg-1.
4. I suggest the authors divide the figure shown in the manuscript into two figures, one summarizing the IL-4/IL-13 signaling pathway and the other summarizing potential cellular effects of IL-4 and IL-13 on inflammatory and structural cells in asthma.

---

## Round 0.2 · accepted · Accept

Congratulations on a good job of correcting your manuscript into an acceptable form.

·

Basic reporting

The authors have significantly improved this review manuscript with the additional information on clinical significance of IL-4/IL-13 pathway, IL-4/IL-13's involvement in the inflammatory response of AR and RA. The new figure 2 is a great summary of IL-4/IL-13 signaling pathway's multiple functions.

I would like to applaud the authors for doing a great job at writing this insightful review manuscript on IL-4/IL-13 and driving the field forward.

Experimental design

N/A

Validity of the findings

N/A

Additional comments

N/A

·

Basic reporting

The paper is well structured and written, with clear objectives.

Experimental design

The paper is technically sound.

Validity of the findings

The paper contributes some new ideas.